# Mobilization of Circulating Tumor Cells after Short- and Long-Term FOLFIRINOX and GEM/nab-PTX Chemotherapy in Xenograft Mouse Models of Human Pancreatic Cancer

**DOI:** 10.3390/cancers15225482

**Published:** 2023-11-20

**Authors:** Yukako Ito, Shinji Kobuchi, Amiri Kawakita, Kazuki Tosaka, Yume Matsunaga, Shoma Yoshioka, Shizuka Jonan, Kikuko Amagase, Katsunori Hashimoto, Mitsuro Kanda, Takuya Saito, Hayao Nakanishi

**Affiliations:** 1Department of Pharmacokinetics, Kyoto Pharmaceutical University, Kyoto 607-8414, Japan; kobuchi@mb.kyoto-phu.ac.jp (S.K.); ky18096@ms.kyoto-phu.ac.jp (A.K.); ky18244@ms.kyoto-phu.ac.jp (K.T.); ky19325@ms.kyoto-phu.ac.jp (Y.M.); ky19383@ms.kyoto-phu.ac.jp (S.Y.); 2Department of Pharmacy, College of Pharmaceutical Sciences, Ritsumeikan University, Shiga 603-8577, Japan; gr0476fr@ed.ritsumei.ac.jp (S.J.); amagase@fc.ritsumei.ac.jp (K.A.); 3Department of Medical Technology, Faculty of Medical Sciences, Shubun University, Ichinomiya City 491-0938, Japan; hashimoto.k@shubun.ac.jp; 4Department of Gastroenterological Surgery, Nagoya University Graduate School of Medicine, Nagoya 466-8550, Japan; m-kanda@med.nagoya-u.ac.jp; 5Department of Gastroenterological Surgery, Aichi Medical University, Nagakute City 480-1195, Japan; saitou.takuya.856@mail.aichi-med-u.ac.jp (T.S.); hnakanis@aichi-med-u.ac.jp (H.N.); 6Laboratory of Pathology, Okazaki City Hospital, Okazaki 444-0002, Japan

**Keywords:** circulating tumor cells, cell and tissue dynamics, chemotherapy, pancreatic cancer, xenograft mouse model, preclinical study

## Abstract

**Simple Summary:**

To date, there has been no definite experimental evidence for the significance of circulating tumor cells (CTCs) as an indicator to estimate the chemotherapeutic effect in cancer patients. We previously reported a transient increase in CTC number 1 week after single-dose chemotherapy in human breast, lung, and gastric cancer xenograft mouse models. In the present study, we extensively examined the dynamics of both CTCs and tumor tissues after single and repetitive doses of chemotherapy in two human pancreatic cancer xenograft models to understand the mechanistic aspect of CTC mobilization after chemotherapy. We confirmed the transient increase in CTC number 1–2 weeks after chemotherapy and proposed a hypothesis that transient CTC mobilization after chemotherapy occurs by the shedding of growth-arrested tumor cells into disrupted tumor blood vessels in the primary tumor tissues 1–2 weeks after chemotherapy, which corresponds to the maximal destructive phase of primary tumor tissues before tissue repair and regeneration.

**Abstract:**

Mobilization of CTCs after various types of therapy, such as radiation therapy, has been reported, but systematic study of CTCs after chemotherapy remained quite limited. In this study, we sequentially examined CTC numbers after single-dose and repetitive-dose chemotherapy, including FORFIRINOX (FFX) and Gemcitabine and nab-Paclitaxel (GnP) using two pancreatic cancer xenograft models. CTC was detected by the immunocytology-based microfluidic platform. We further examined the dynamic change in the histology of primary tumor tissues during chemotherapy. We confirmed a transient increase in CTCs 1–2 weeks after single-dose and repetitive-dose of FFX/GnP chemotherapy. Histological examination of the primary tumors revealed that the peak period of CTC at 1–2 weeks after chemotherapy corresponded to the maximal destructive phase consisting of cell cycle arrest, apoptosis of tumor cells, and blood vessel destruction without secondary reparative tissue reactions and regeneration of tumor cells. These findings indicate that mobilization of CTCs early after chemotherapy is mediated by the shedding of degenerated tumor cells into the disrupted blood vessels driven by the pure destructive histological changes in primary tumor tissues. These results suggest that sequential CTC monitoring during chemotherapy can be a useful liquid biopsy diagnostic tool to predict tumor chemosensitivity and resistance in preclinical and clinical settings.

## 1. Introduction

Circulating tumor cells (CTC) are considered potential biomarkers for providing diagnostic and therapeutic solutions in various types of cancers. They offer a minimally invasive and easily reproducible method for detecting tumor cells and monitoring changes in tumor cell numbers and genetic alterations before and after various types of therapy [1,2]. Therefore, CTCs have multimodal potential as early diagnostic and prognostic markers, as well as predictive markers for therapeutic responses in various types of malignancies [3,4,5]. Concerning the response of CTCs to cancer therapy, we previously reported that CTCs transiently increased 1 week after drug therapy in human breast, lung, and gastric cancer xenograft mouse models. Such a transient increase in CTCs shortly after chemotherapy is the result of tumor cell mobilization from the primary tumor into the blood and is a potential indicator for monitoring therapeutic response [6,7]. 

Pancreatic cancer (PC) is the fourth leading cause of cancer-related deaths in Asia, the USA, and Europe [8]. Surgical resection is the only method for long-term survival in PC patients, but the median survival of patients undergoing curative pancreatectomy alone is 18–20 months, with a 5-year survival rate of 10% [9]. Chemotherapy, including adjuvant and neoadjuvant therapies, improves the median and 5-year overall survival (OS) of patients with curatively resected PC [10]. Neoadjuvant chemotherapy (NAC) using agents such as FOLFIRINOX and GEM/nab-PTX offers several advantages over upfront surgery, including the delivery of systemic chemotherapy to almost all patients treated with surgery and a higher negative-margin resection rate, leading to improved OS [11,12]. Although treatment modalities for pancreatic cancer have progressed, the survival of patients with pancreatic cancer remains poor. This is mainly because of the difficulty in early diagnosis by imaging and the biologically aggressive nature of pancreatic cancer, which is characterized by high levels of hematogenous and lymph node metastases from an early stage [13]. 

Several clinical studies on pancreatic cancer have shown that high levels of CTCs are associated with tumor progression and are correlated with a patient’s short survival, indicating the utility of CTCs as prognostic markers [14]. However, there are few reports on the effectiveness of CTCs as indicators of therapeutic response in patients with pancreatic cancer. Possible reasons for this are the little clinical experience in the detection of CTC during NAC in pancreatic cancer [15] and the lack of a mouse pancreatic cancer CTC model that allows the examination of CTCs for monitoring therapeutic responses. 

We recently established a new mouse CTC model bearing human pancreatic adenocarcinoma cells (SUIT-2) for preclinical studies that can sequentially detect CTCs after chemotherapy using an immunocytology-based CTC detection platform developed in our laboratory [16]. In the present study, using two mouse pancreatic cancer CTC models (SUIT-2 and KP-4) [17,18], we extensively examined both CTC and tissue dynamics of the primary tumor in response to short- and long-term repetitive chemotherapy to understand the dynamics and mechanism of CTC mobilization after chemotherapy.

## 2. Materials and Methods

### 2.1. Reagents

Rabbit polyclonal antibodies against a wide spectrum of human cytokeratin (CK-wide) (ab9377) and rabbit monoclonal antibodies against CD31 (ab182981) were purchased from Abcam (Cambridge, UK). The FOLFIRINOX regimen consisted of 5-fluorouracil (5-FU), leucovorin, and irinotecan obtained from Fujifilm Wako Chemical (Tokyo, Japan), and oxaliplatin (Elplat^®^) obtained from Yakult Honsha Co., Ltd. (Tokyo, Japan). The GnP therapy consisted of gemcitabine and nab-paclitaxel. Gemcitabine was purchased from Fujifilm Wako Chemical, and nab-paclitaxel (Abraxan^®^) was purchased from TAIHO PHARMACEUTICAL Co., Ltd. (Tokyo, Japan).

### 2.2. Cell Lines

The SUIT-2 cell line was obtained from the JCRB Cell Bank (Japanese Collection of Research Bioresources Cell Bank, Osaka, Japan), which was established from liver metastases of Japanese patients with pancreatic cancer [17]. Another pancreatic cancer cell line (KP-4) was also obtained from the JCRB Cell Bank [18]. These cell lines were maintained in Dulbecco’s modified Eagle’s medium containing 10% fetal bovine serum (GIBCO, Grands Islands, NY, USA) with 100 units/mL penicillin and 100 units/mL streptomycin sulfate and cultured in a humidified 5% CO_2_ incubator at 37 °C.

### 2.3. Animals

Seven- to nine-week-old female athymic nude mice (KSN strain; 23–25 g) were obtained from Japan SLC (Hamamatsu, Japan). The mice were maintained under specific pathogen-free (SPF) conditions. All animal experiments were performed according to an experimental protocol approved by the Ethics Review Committee for Animal Experimentation of Kyoto Pharmaceutical University (approval number: PKPD-18-002) and met the standards defined by recently reported international guidelines [19].

### 2.4. Animal Experiment Using Pancreatic Cancer Xenograft Mouse CTC Models

Based on a previous report [16], in this study, we used a subcutaneous (heterotopic) transplantation model rather than an orthotopic (pancreatic) surgical transplantation model because of the higher number of CTCs, less local surgical stress, and no risk of fatal peritoneal metastasis in the subcutaneous model.

SUIT-2 cells and KP-4 cells (3 × 10^6^/0.1 mL) in HBSS were subcutaneously (sc) injected into two sites in the back region of the mice using a 27G syringe. At approximately 1 month post-sc injection, mouse blood before treatment was collected by cardiac puncture. The mice received FFX or GnP chemotherapy or vehicle (saline) via the tail vein. The FFX solution (1 mL) was prepared by combining calcium folinate (10 mg), irinotecan (5 mg), and 5-fluorouracil (5 mg) with 0.1 mL of oxaliplatin (Elplat^®^, 5 mg/mL) and then dissolving them in a 5% glucose solution (0.9 mL). The prepared FFX solution (0.1 mL/20 g mice) was administered into the caudal vein; the final dosages were calcium folinate (50 mg/kg), oxaliplatin (2.5 mg/kg), irinotecan (25 mg/kg), and 5-fluorouracil (25 mg/kg) twice weekly for 7 weeks. The GnP solution was prepared by combining GEM (20 mg) with nab-paclitaxel (Abraxane^®^, 2 mg) and dissolving it in saline (1 mL). The prepared solution (0.1 mL/20 g mice) was administered into the caudal vein; the final dosages were GEM (100 mg/kg) and nab-PTX (10 mg/kg). The dose administered to the mice was determined once weekly for 7 weeks based on previously reported animal experiments [20]. Control animal experiments were performed with the administration of 5% glucose solution for FFX and saline for the GnP regimen.

### 2.5. Blood Sampling

Blood was collected by cardiac puncture in a specialized tube (Streck, La Vista, NE, USA) for liquid biopsy, including CTC and cfDNA, using a 27G syringe under sedation with isoflurane. The blood sampling route was determined as a result of preliminary experiments, including the lateral tail vein, retroorbital venous plexus, and heart [5]. The cardiac puncture has advantages in terms of safety, reproducibility, and virtually no risk of contamination of normal epithelial cells. The volume of blood sampled was based on body weight and interval period, as recommended by animal guidelines [19]. Usually, we collected approximately 0.20 mL of blood per mouse.

### 2.6. CTC Isolation from Mouse Blood by Filtration-Based Microfluidic Platform

CTCs were detected principally using a previously reported method [21]. Briefly, it consists of the following three steps: (1) enrichment of CTCs using an automated CTC collection apparatus with a constant fluid pressure system (Maruyasu Industry Co., Okazaki, Japan) and a 3-dimensional (3D) metal filter device (Optnics Precision Co., Tochigi, Japan), (2) transfer of enriched CTCs from the filter to a glass slide (CTC glass slide) using an air pressure-mediated automated CTC transfer apparatus (Maruyasu Industry Co.), and (3) cytological examination of the CTC glass slide using cytokeratin immunocytochemistry (ICC) and subsequent nuclear counterstaining with hematoxylin.

Mouse blood (0.2 mL) was diluted 50-fold with PBS containing 5 mM EDTA (PBS/EDTA) and then filtered with the CTC collection device described above. After filtration, enriched CTCs trapped in the filter were fixed with 10% formalin for 30 min and washed with PBS/EDTA in the device. The filter, detached from the device, was placed upside down on a coated glass slide (MAS coat, Matsunami, Osaka, Japan), and the CTCs trapped in the filter were quickly transferred to a glass slide using an automated transferring device. The resulting CTCs attached to the glass slide were immediately fixed in 95% ethanol at 4 °C for preservation, followed by a further 10 min of buffered formalin fixation before immunocytochemistry.

### 2.7. Immunocytological and Cytological Staining of CTCs

For immunostaining, after treatment with peroxidase and a protein-blocking reagent, the specimens were incubated with rabbit polyclonal anti-cytokeratin antibody (CK-wide) for 1 h. After washing, the specimens were incubated with an HRP-labelled polymer conjugated to a goat anti-rabbit antibody (EnVision+System) (DAKO, Carpinteria, CA, USA) for 30 min. After washing with PBS, the chromogen was developed using a Liquid DAB+substrate chromogen system (DAKO). Nuclei were counterstained with Meyer’s hematoxylin. To estimate the conservation of cell morphology, Papanicolaou (Pap) staining of CTC glass slides was performed using an automatic stainer (Sakura Fintec, Tokyo, Japan). The CTCs were observed under a light microscope (Olympus BX50, Tokyo, Japan).

### 2.8. Histological and Immunohistochemical Staining of Primary Tumor Tissues

The subcutaneous mouse tumors were resected and fixed with 10% buffered formalin for 24–48 h, embedded in paraffin, and sectioned into 5 μm slices. Hematoxylin and eosin staining was performed using a routine method. Immunohistochemical staining (IHC) of CD31 was performed as follows: briefly, the sections were pre-treated with microwave heating at 98 °C for 15 min (PH9.0), then blocked for peroxidase and protein on the tissue. The tissues were then incubated with a primary antibody against human CD31 for 2 h at room temperature. After washing, the specimens were incubated with an HRP-labelled polymer conjugated to a goat anti-rabbit antibody (EnVision+System) (DAKO, Carpinteria, CA, USA) for 30 min. The chromogen was developed using the Liquid DAB+substrate chromogen system (DAKO). Nuclei were counterstained with Meyer’s hematoxylin.

### 2.9. Statistical Analyses

All values are expressed as the mean ± S.D. Differences were assumed to be statistically significant at *p* < 0.05 for Student’s unpaired *t*-test and/or Mann–Whitney U-test or Bonferroni’s multiple comparison test.

## 3. Results

### 3.1. Sequential Changes in CTCs after Single-Dose FFX and GnP Chemotherapy in the SUIT-2 Mouse Tumor Models

Atypical cells with keratin-positive cytoplasm and hematoxylin-stained enlarged nuclei on glass slides were considered CTCs under light microscopy. These CTCs were observed in both single-cell and small or large-cell clusters. The number of CTCs significantly increased 4–7 days after FFX chemotherapy compared with before treatment (*p* < 0.01) and decreased 14 days after chemotherapy. This transient increase in CTCs was not observed after treatment with the vehicle: 5% glucose solution. Interestingly, the increased CTCs at 4–7 days after FFX chemotherapy were mainly composed of cluster-type CTCs (Figure 1A,B). A similar transient increase in CTC numbers after chemotherapy was observed 7 days after GnP chemotherapy (*p* < 0.01) (Figure 1C,D). The maximal increase in CTC numbers 7 days after chemotherapy was higher in FFX than GnP therapy, although it is not significant.

### 3.2. Histological Changes in the Primary Tumor Tissues(SUIT-2) after Single-Dose FFX and GnP Chemotherapy

Histological examination demonstrated severe growth arrest and apoptosis induction in primary tumor tissues 4–7 days after FFX chemotherapy (Figure 2A). The mitotic index (mitotic tumor cell number/high-power field) significantly decreased 4–7 days vs. day 0 after FFX chemotherapy (*p* < 0.01, vs. day 0) (Figure 2B). This decrease in the mitosis of tumor cells is in sharp contrast to the transient increase in CTC numbers after FFX chemotherapy. Apoptosis was also induced 4–7 days after FFX therapy as a tendency (Figure 2D). In addition, CD31 immunostaining showed that tumor blood vessels were severely damaged, such as the destruction of endothelial cell-cell interactions, which increased the chance of intravasation of tumor cells into blood vessels and subsequent thrombus formation (Figure 2C).

Similar severe disruption of primary tumor tissue, including mitotic arrest, apoptosis induction of tumor cells, and breakage of tumor blood vessels, were also observed 4–7 days after GnP chemotherapy (Figure 3A,B,D). Furthermore, CD31 staining revealed that destruction of endothelial cell-cell junction and subsequent intravasation of tumor cells into blood vessels were observed in the primary tumor tissues 4–7 days after GnP chemotherapy (Figure 3C).

### 3.3. Sequential Changes in CTCs and Histology of the Primary Tumor Tissues after Single-Dose FFX Chemotherapy in the KP-4 Mouse Tumor Model

Like the SUIT-2 model, CTCs were observed mainly in small or large-cell clusters with some single cells in the KP-4 mouse tumor model (Figure 4A). The number of CTCs significantly increased 7–14 days after FFX chemotherapy and decreased up to 21 days after chemotherapy (Figure 4B), indicating a slight time shift of CTC peak from 7 days in the SUIT-2 model to 11 days in the KP-4 model. Histological examination demonstrated severe growth arrest due to mitotic arrest and apoptosis induction in primary tumor tissues during 7–14 days after FFX chemotherapy (Figure 4C,D). This decrease in the mitotic tumor cells contrasts with the increase in the CTC numbers after single-dose FFX chemotherapy, indicating synchronization of CTC increase and growth arrest of the primary tumor. This suggests an intimate relationship between CTC mobilization into the blood circulation and destructive changes in primary tumor tissue.

### 3.4. Sequential Changes in the CTC Numbers after Long-Term, Repetitive FFX and GnP Chemotherapy

FFX was administered twice weekly for 7 weeks, and the tumor volume was monitored sequentially (Figure 5A). FFX was highly effective, and the tumor volume markedly reduced after therapy (*p* < 0.001 vs. control). Blood samples were collected from week 0, early stage (1–2 weeks), middle stage (3–4 weeks), and late stage (5–7 weeks) after chemotherapy, and CTC numbers were sequentially examined. Similar to the single-dose FFX treatment, CTC numbers highly increased 1–2 weeks and lowly increased 3–4 weeks after FFX therapy (*p* < 0.05, vs. control) and decreased thereafter (5–7 weeks) (Figure 5B).

GnP was administered once a week for 7 weeks, and tumor volume was monitored during this period. Tumor volume was significantly reduced (*p* < 0.005) after long-term, repetitive GnP chemotherapy. The antitumor activity of GnP was significantly lower than that of FFX (*p* < 0.05) (Figure 5C). Similar to the single-dose GnP treatment, the CTC numbers significantly increased 1–2 weeks after GnP therapy (*p* < 0.05 vs. control) and decreased thereafter during 3–7 weeks, which was comparable with the non-treatment control (Figure 5D). A comparison of the effects of FFX and GnP chemotherapy on the increase in the CTC number revealed that FFX was significantly stronger than GnP (*p* < 0.05), similar to the antitumor activities of these two drugs.

### 3.5. Histological Changes in Primary Tumor Tissues after Long-Term FFX and GnP Chemotherapy

Histologically, pure and strong destructive changes, such as maximal mitotic arrest, apoptosis induction, and breakage of tumor blood vessels, were observed only in the early stage (1–2 weeks) after FFX or GnP chemotherapy, as shown in the case of single-dose chemotherapy (Figure 2A and Figure 3A). During the middle (3–4 weeks) and late stage (5–7 weeks) after repeated chemotherapy, CTCs are mostly composed of single cells and small clusters (Figure 6A). On the other hand, tumor tissues become heterogeneous with a mixed population consisting of (1) highly atypical, degenerative tumor cell areas with regenerated blood vessels, reparative fibrous scar, and (2) regrowing tumor cell areas with high mitotic activity (Figure 6B,C). In these tissues, disruption of blood vessels and tumor cell intravasation were rarely observed. In contrast, the primary tumor tissue of the control mice before chemotherapy was homogenous, consisting of small tumor cells with high mitotic figures and almost intact blood vessels (Day 0 in Figure 2A and Figure 3A).

## 4. Discussion

We previously reported a transient increase in CTCs after single-dose chemotherapy, a unique pathobiological phenomenon, in three mouse xenograft models, including human breast, lung, and gastric cancers [6,7]. In the present study, to understand the reason this CTC increase occurs shortly after chemotherapy, we extensively investigated both CTC and tumor tissue dynamics after single-dose and long-term, repeat-dose chemotherapy using two pancreatic cancer mouse CTC models.

The following four conclusive findings were drawn: (1) We confirmed a transient increase in CTCs consisting mainly of cluster-type CTCs 1–2 weeks after single-dose chemotherapy, irrespective of whether FFX or GnP regimen and SUIT-2 or KP-4 pancreatic cancer models. FFX chemotherapy showed stronger tumor growth inhibitory activity than GnP therapy (*p* < 0.05). Consistent with such antitumor activity of the drug, the peak number of CTCs 1–2 weeks after single-dose and repetitive-dose FFX therapy was significantly higher than that 1–2 weeks after GnP therapy (*p* < 0.05), suggesting that the extent of transient increase in the number of CTCs correlates with the antitumor activity of the drug. (2) In long-term, repetitive chemotherapy experiments, we found that clear CTC increase was evident in only the early phase (1–2 weeks) after chemotherapy, irrespective of the FFX or GnP, like the single-dose chemotherapy experiment. During the middle to late stages after chemotherapy, CTC numbers were only weakly increased 3–4 weeks after FFX chemotherapy with stronger antitumor activity than GnP. (3) Histological examination of the primary tumor after single-dose chemotherapy demonstrated maximal mitotic arrest and apoptosis induction of tumor cells in the primary tumor tissue 1–2 weeks after chemotherapy, indicating the occurrence of complete cell cycle arrest at the G1/S and G2/M phases. CD31 immunostaining further demonstrated breakage of tumor blood vessels and intravasation of tumor cells into the blood vessels 1–2 weeks after chemotherapy. (4) Histological examination of the primary tumor after repetitive drug therapy demonstrated that the number of viable tumor cells in the tumor tissue constantly decreased with the progression of chemotherapy. However, regrowing tumor cells appeared from 2 to 3 weeks after the first dose of chemotherapy and damaged tumor blood vessels were also regenerated with reparative fibrous scar tissue, indicating continuous recovery and remodeling of the tumor tissue during repetitive drug therapy. Therefore, the primary tumor tissue from middle stage to late stage after repeated chemotherapy changed a complex and heterogeneous histological architecture consisting of regrowing tumor cells and reparative tumor stroma. This is in sharp contrast to the simple and pure destructive tumor tissue without any secondary tissue reactions 1–2 weeks after single-dose chemotherapy.

Interestingly, the clear increase in CTC number after repetitive chemotherapy substantially occurred only 1–2 weeks after chemotherapy, similar to the single-dose chemotherapy experiment. The early phase (1–2 weeks) after chemotherapy is a histologically unique time point of primary tumor tissue with the purest destructive phase without significant secondary histological reactions such as regeneration of tumor cells, repair of blood vessels, and formation of fibrous scar. These results strongly suggest that increased mobilization of tumor cells from the primary tumor to the blood occurs primarily through passive intravasation of tumor cells as a result of fatal destruction of tumor cell-cell interactions and endothelial cell-cell interactions of tumor blood vessels, in addition to apoptosis and necrosis in the primary tumor tissue.

In other words, transient CTC mobilization after chemotherapy is not a physiological or pathological event but an artificial event driven by the purest tissue disruption associated with chemotherapy. This idea of passive intravasation contrasts with active intravasation, which is driven by the increased invasive activity of the tumor cells themselves, observed in normal tumor tissues without therapy. In this respect, it is of note that G2/M-arrested cells produced by chemotherapy are round-shaped cells with reduced cell-cell and cell-extracellular substratum attachments. Therefore, the possibility cannot be excluded that G2/M-arrested tumor cells increased their motility and actively intravasated into blood vessels early after chemotherapy [6]. Based on the findings described above (1)–(4), we propose a hypothesis on the mechanistic aspect of the CTC mobilization in the early phase (1–2 weeks) after single and repetitive chemotherapy. In the highly disrupted primary tumor tissue, fragmented but still viable tumor cell clusters that survived after the severe cytocidal effect of chemotherapy are ejected by the histological pressure as the result of tissue disruption and are shed into the blood through breakage of blood vessels. Therefore, the transient increase in the number of CTCs early after chemotherapy is dependent on the degree of disruption of tissue architecture caused by chemotherapy. In this respect, it is reasonable to assume that during the middle to late stage after chemotherapy, tumor tissue is reconstituted by regrowing tumor cells and regenerating blood vessels supported by the reparative fibrous tissue. Tumor tissues are now reconstituted and act as a barrier to protect tumor cells from shedding and subsequent mobilization into the blood circulation.

In the past, several investigators have reported that surgical intervention in a variety of cancer patients is associated with the mobilization of CTCs after surgical manipulation [22]. More recently, Martin et al. and several other groups reported the mobilization of CTCs into the peripheral circulation after radiation therapy [23,24]. The mobilization of CTCs into peripheral circulation by chemotherapy has been investigated in several mouse models, including our previous models [6,7]. Inhestern et al. reported sequential changes in CTC numbers before and after chemotherapy in patients with oral cancer [25]. However, reports on CTC mobilization by antitumor drug therapy in clinical cancer patients are still quite limited, except for the Ortiz–Otero report [26,27]. Our present systematic experimental report with serial monitoring of CTC numbers before and after chemotherapy using multiple mouse CTC models is, to our knowledge, the first report demonstrating the mechanistic aspect of CTC mobilization after chemotherapy.

## 5. Conclusions

We extensively examined the acute and chronic dynamics of CTCs after short- and long-term chemotherapy in two mouse pancreatic cancer models. The number of CTCs mobilized early after chemotherapy reflects the therapeutic effects of chemotherapy on primary tumors. These results suggest that sequential monitoring of CTCs in terms of number and morphology (single or cluster) before and after chemotherapy using our immunocytology-based CTC detection platform could be a potentially powerful diagnostic tool for liquid biopsy to predict the response to anti-cancer therapy in various clinical settings. Further studies, including the molecular mechanism of CTC mobilization and the effects of mobilized CTCs on the metastatic potential after chemotherapy, are now planned in the following study.

## Figures and Tables

**Figure 1 cancers-15-05482-f001:**
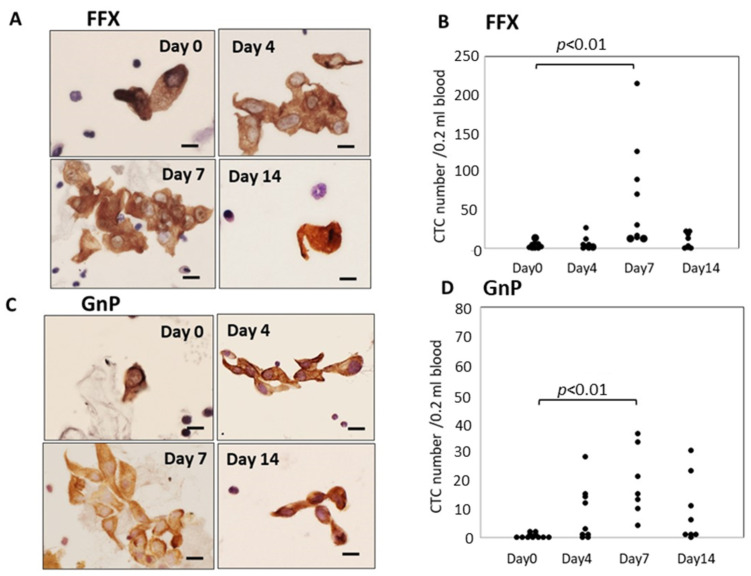
Sequential changes in the number and morphology of CTCs after single-dose FFX and GnP chemotherapy in Suit-2 pancreatic cancer xenograft mice. (**A**). Cytokeratin-positive CTCs in blood obtained at 0, 4, 7, and 14 days after FFX treatment. Cluster-type CTCs are often seen 4 and 7 days post-administration. Bar = 20 μm. (**B**). Sequential changes in CTC numbers (average) according to days 0 (2.25), days 4 (6.86), days 7 (71.9), and days 14 (9.88) after FFX treatment, respectively. Significant transient increase in the number of CTC is seen on day 7 vs. day 0 (*p* < 0.01). (*n* = 7–8). (**C**). Cytokeratin-positive CTCs in blood obtained at 0, 4, 7, and 14 days after GnP treatment. Cluster-type CTCs are often seen 4 and 7 days post-administration. Bar = 20 μm. (**D**). Changes in CTC numbers (average) according to days 0 (0.55), day 4 (8.22), day 7 (18.9), and 14 days (9.13) after GnP treatment, respectively. Significant transient increase in the number of CTC is seen on day 7 vs. day 0 (*p* < 0.01). (*n* = 7–8).

**Figure 2 cancers-15-05482-f002:**
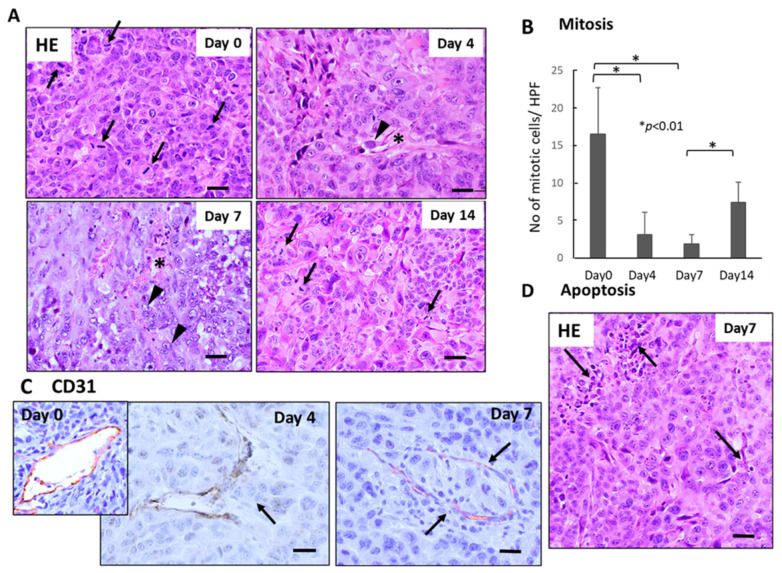
Sequential changes in the histology of primary tumor tissues after single-dose FFX chemotherapy in Suit-2 tumor-bearing mice (*n* = 5). (**A**) Hematoxylin-Eosin (HE) staining of primary tumor tissues obtained at 0, 4, 7, and 14 days after FFX treatment. Black arrows and arrowheads indicate mitotic and intravasating tumor cells, respectively. Asterisk * indicates damaged blood vessels. Bar = 30 µm. (**B**) Sequential analysis of mitotic index; mitotic tumor cell numbers/high-power fields (HPF) according to days (0, 4, 7, and 14 days) after FFX treatment. Significant decrease of mitosis in the tumor cells on day 4 and day 7 after therapy (* *p* < 0.001, vs. day 0) is apparent. (**C**) CD31 immunostaining of primary tumor tissues was obtained at 4 and 7 days after FFX treatment. In contrast to day 0, intravasation of tumor cells into endothelium-damaged tumor blood vessels and tumor thrombus formation in the blood vessel is seen 4 and 7 days after FFX therapy (arrows). Bar = 30 µm. (**D**) Apoptotic tumor cells and clusters (arrows) are seen in the non-necrotic area of the tumor tissues 7 days after FFX therapy. Bar = 30 µm.

**Figure 3 cancers-15-05482-f003:**
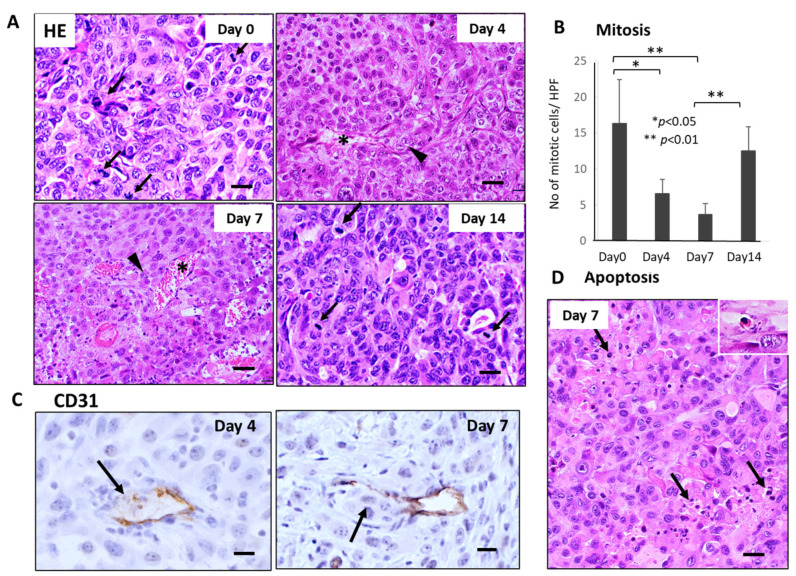
Sequential changes in the histology of primary tumor tissues after single-dose GnP chemotherapy in Suit-2 tumor-bearing mice (*n* = 4). (**A**) Hematoxylin-Eosin (HE) stained primary tumor tissues obtained at 0, 4, 7, and 14 days after single-shot GnP treatment. Black arrows and arrowheads indicate mitotic and intravasating tumor cells, respectively. Asterisk * indicates damaged blood vessels. Bar = 30 µm. (**B**) Sequential analysis of mitotic index; numbers of mitotic tumor cells /HPF according to days (0, 4, 7, and 14 days) after GnP treatment. Significant decrease of mitosis in the tumor cells at day 4 and day 7 after therapy is apparent (** *p* < 0.001, vs. day 0). Bar = 30 µm. (**C**) CD31 immunostaining of primary tumor tissues obtained at 4 and 7 days after GnP treatment. Breakage of endothelial cell-cell junction and intravasation of tumor cells into the blood vessels are seen (arrows). (**D**) Apoptotic tumor cells and clusters (arrows) are often seen in the non-necrotic area of the tumor tissues. Inset: enlarged view of apoptotic tumor cell. Bar = 30 µm.

**Figure 4 cancers-15-05482-f004:**
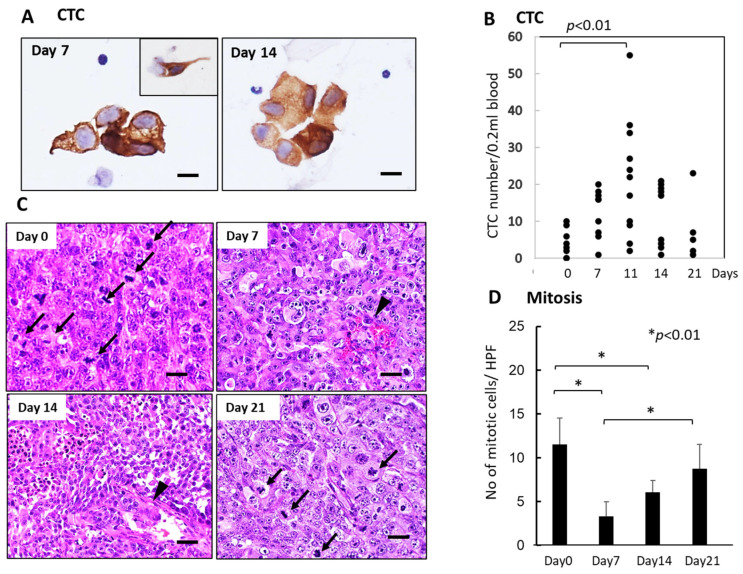
Sequential changes in the number of CTCs in blood and histology of the primary tumor tissues after single-dose FFX chemotherapy in KP-4 pancreatic cancer xenograft mice (*n* = 4). (**A**) Cytokeratin-positive CTCs in blood obtained at 7 and 14 days after FFX treatment. Small cluster-type CTCs and single-cell type CTCs (inset) are seen 7 and 14 days post-administration. Bar = 30 μm. (**B**) Sequential analysis of CTC average numbers is performed according to day 0 (4.25), day 7 (12.3), day 11 (21.8), day 14 (12.0), and day 21 (7.60) after FFX treatment, respectively. Significant increase in the number of CTC on day 11 vs day 0 (*p* < 0.01) is seen. (*n* = 8–11) (**C**) Sequential changes in the histology of primary tumor tissues after single-dose FFX chemotherapy. Hematoxylin-Eosin (HE) staining of the primary tumor tissues obtained at 0, 7, 14, and 21 days after FFX treatment. Black arrows and arrowheads indicate mitotic and intravasating tumor cells, respectively. Bar = 20 µm. (**D**) Quantitative analysis of mitotic tumor cell numbers/high-power fields (HPF) according to days (0, 7, 14, and 21 days) after FFX treatment. Significant decrease of mitotic tumor cells at day 7 and day 14 after therapy (* *p* < 0.01, vs. day 0) is apparent. Bar = 30 µm Bar = 30 µm.

**Figure 5 cancers-15-05482-f005:**
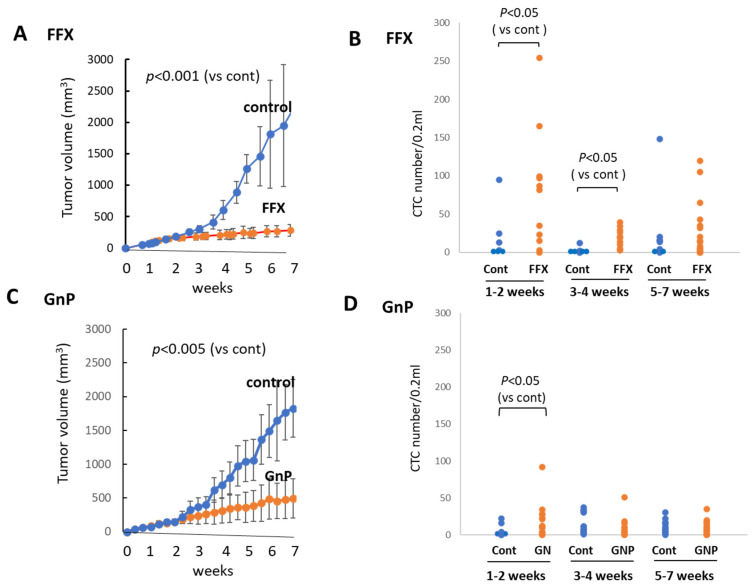
Sequential changes in tumor growth and CTC numbers during long-term, repetitive FFX and GnP chemotherapy in Suit-2 tumor-bearing mice. (**A**) Growth inhibition of primary tumors with FFX therapy (*p* < 0.001, vs. control). (**B**) Sequential changes in CTC numbers (average) during early (1–2 weeks), middle (3–4 weeks), and late stage (5–7 weeks) after repetitive FFX treatment *(n* = 5–19). Average CTC numbers (Cont, FFX) are 19.3, 78.1 (early), 3.5, 19.9 (middle), and 31.1, 23.5 (late), respectively. Significant increase in CTC number is observed during the early stage and middle stage (*p* < 0.05, vs. control). (**C**) Growth inhibition of primary tumors with GnP therapy (*p* < 0.005, vs. control). (**D**) Sequential changes in CTC numbers (average) during early (1–2 weeks), middle (3–4 weeks), and late stage (5–7 weeks) cha–7 weeks) after repetitive GnP treatment (*n* = 7–16). Average CTC numbers (Cont, GnP) are 4.17, 17.5 (early), 14.0, 8.73 (middle), and 9.81, 8.26 (late), respectively. CTC is significantly increased only in the early stage after GnP therapy (*p* < 0.05 vs. control).

**Figure 6 cancers-15-05482-f006:**
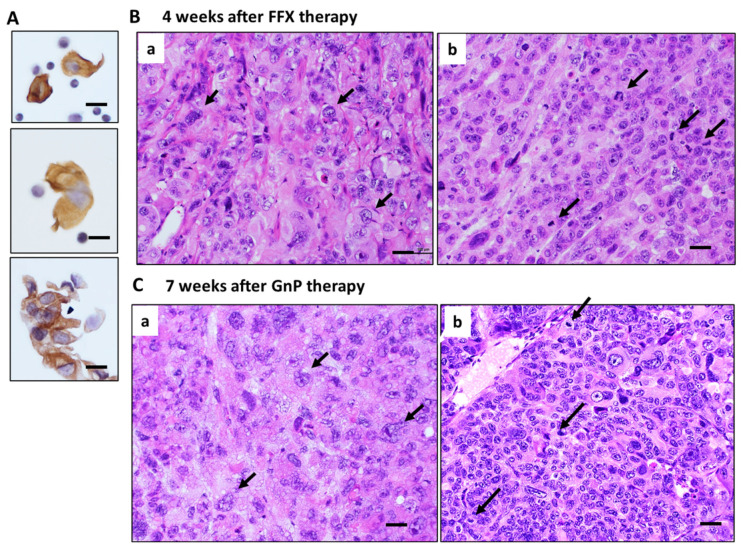
Histology of primary tumors during middle to late stage after long-term, repetitive FFX and GnP therapy in Suit-2 tumor-bearing mice. (**A**) CTCs observed during the middle and late stages after chemotherapy. Single-cell and small cluster-type CTCs are seen. Bar = 20 µm. (**B**) Tumor tissue 4 weeks after long-term FFX therapy is heterogenous, consisting of large, atypical tumor cell area (black arrows with few mitotic figures (**a**) and atypical small tumor cells with high mitotic figures (black arrows) (**b**). (**C**) Tumor tissue 7 weeks after long-term GnP therapy is also heterogeneous, consisting of large, atypical tumor cells (black arrows) with low mitotic figures (**a**) and small, atypical tumor cells with high mitotic figures (black arrows) (**b**). Bar = 30 µm.

## Data Availability

Data are contained within this article.

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
