# Peer review of "Mobilization of Circulating Tumor Cells after Short- and Long-Term FOLFIRINOX and GEM/nab-PTX Chemotherapy in Xenograft Mouse Models of Human Pancreatic Cancer"

_cancers, 2023, doi:10.3390/cancers15225482_

Round 1

Reviewer 1 Report

Comments and Suggestions for Authors

The authors studied the effects of single and repeated doses of chemotherapy on the number of CTCS and suggested the reason for the transient mobilization of CTCS after chemotherapy. The content of the article has scientific research value, but there are some questions as follows.

1. The references cited in the article are few, please add more recent references

2. The Sequential changes of CTCs after single-dose FFX and GnP chemotherapy in the SUIT-2 mouse tumor models were presented in this paper, but no further theoretical analysis of this phenomenon was conducted.

3.Histological changes of the primary tumor tissues(SUIT-2) after single-dose FFX and GnP chemotherapy just describe the observed phenomena, No further analysis was performed.

4. Why were the observation results recorded on days 0, 4, 7 and 14 chosen? Were the number of experimental groups selected too little?Whether the conclusion has particularity?

Author Response

Thank you for the helpful comments to clarify and strengthen our manuscript.

We answer these questions one by one as follows.

Reviewer 1 comment

  1. The references cited in the article are few, please add more recent references.

Answer; According to the reviewer’s suggestions, we deleted 2 old CTC papers and added 5 more recent references (mostly 2022-2023) in the introduction section and in the discussion section.

  1. The Sequential changes of CTCs after……. but no further theoretical analysis of this phenomenon was conducted.
  2. Histological changes of the primary tumor tissues (SUIT-2) after single-dose FFX and GnP chemotherapy just describe the observed phenomena, No further analysis was performed.

Answer; Thank you for valuable comments (2 and 3). Since these comments seem to be related questions, we answer these questions together as follows; Present study is, to our knowledge, the first in vivo study to reveal mechanistic aspect of CTC mobilization after chemotherapy in two experimental mouse CTC models. CTC mobilization is not in vitro phenomenon, but in vivo phenomenon originated from primary tumor after chemotherapy. We think the most critical things in vivo study is to clarify the real events occurred in the primary tumor tissues during CTC mobilization after chemotherapy systematically based on the detailed histological analysis. Therefore, in this study, we focused our attention on the sequential histological/pathological events of the primary tumor including tumor cells, tumor blood vessels, and connective tissue stroma after chemotherapy in relation to CTC mobilization using multiple experimental conditions with varying mouse models, drugs and regimens. Once pathological events are established, we are going to proceed next step. 

In this respect, we add following sentence in the “Conclusion section”; Further studies including molecular mechanism of CTC mobilization after chemotherapy and biological significance such as metastatic potential of the mobilized CTCs are planned in the following studies.  

  1. Why were the observation results recorded on days 0, 4, 7 and 14 chosen?

Were the number of experimental groups selected too little?Whether the conclusion has particularity?

Answer; Thank you for your important comment. Based on our previous studies using breast cancer and lung cancer xenograft mouse models (Reff; 6,7), we revealed a transient increase in CTC number during 4-10 days after single-dose chemotherapy. This previous findings are the reasons for the choice of 4-7-14 days after single-dose chemotherapy in this study. In the present study, we confirmed a transient increase in the CTC number during days 4-7 with a peak at days 7 after single-dose chemotherapy with FFX in SUIT-2 model and a transient increase during days 0, 7, 11, 14 and 21 with a peak at days 11 after single-dose chemotherapy with FFX in KP-4 mouse model. This indicates slight difference in the peak time of the transient increase after single-dose chemotherapy depending on the mouse models and drugs used. Furthermore, results from long-term, repetitive experiment also showed a transient increase in the number of CTC at least 1-2 weeks after first-dose chemotherapy also support a transient increase of CTC number after initial dose of the chemotherapy, indicating that a transient increase of CTC is a general phenomenon, but not particular one. According to the important comment of the reviewer, we will further examine CTC number for more longer range (more than 1 month) after single-dose chemotherapy to accumulate our present result in more detail.

Reviewer 2 Report

Comments and Suggestions for Authors

This study presents a paper that emphasizes the importance of initial monitoring of drug responses in pancreatic cancer, by investigating the mobilization of CTCs in relation to two representative chemotherapy agents for pancreatic cancer. It demonstrates highly significant results in the field of liquid biopsy. Additionally, this study approached the experiment systematically through the xenograft method, yielding interesting results.

However, there are several aspects of the experimental data that require important additions and explanations as major revisions.

1.     In Figures 1-4, there are no control results for the two drugs. It seems necessary to compare the CTC results from tissues not treated with drugs.

2.     Although all the results indicate an increase in the number of CTCs, specific numbers are needed. It would be better if the average number of CTCs could be stated in all possible parts of the results section.

3.     In Figure 5, the change in the number of CTCs over the long term is shown. However, for weeks 3-4, the distribution pattern shows a higher number of CTCs compared to the control, and the results for weeks 5-7 appear to be the same. It seems necessary to state the average number of CTCs in the results section, and additional explanation is required if possible.

4.     While the number of circulating tumor cells is important, the expression levels of mutant genes or specific mRNA genes in the separated cells are also critical issues. It would be beneficial to add content on this topic to the Discussion section.

5.     The CTC separation method is based on filtration, which is greatly influenced by the size of the CTCs. It would be good to add content on this topic to the Discussion section as well.

Author Response

Thank you for the helpful comments to clarify and strengthen our manuscript.

We answer these questions one by one as follows.

Reviewer 2 comment

  1. In Figures 1-4, there are no control results for the two drugs. It seems necessary to compare the CTC results from tissues not treated with drugs.

Answer; We carried out control experiment using vehicle (5% glucose solution for FFX) instead of the drug and already described shortly “this transient increase in CTCs was not observed after treatment with the vehicle” in the result section of 3.1 (page 5). We newly added vehicle name, 5% glucose solution and saline in the Materials and Method section of 2.4 (page 3).

  1. Although all the results indicate an increase in the number of CTCs, specific numbers are needed. It would be better if the average number of CTCs could be stated in all possible parts of the results section.

Answer; Following reviewer’s comment, we added the average CTC numbers in all possible parts of the results section in the Figure legends (Figure 1B, 1D and Figure 4B and Figure 5B, 5D).

  1. In Figure 5, change in the number of CTCs over the long term is shown. However, for weeks 3-4, the distribution pattern shows a higher number of CTCs compared to the control, and the results for weeks 5-7 appear to be the same. It seems necessary to state the average number of CTCs in the results section, and additional explanation is required if possible.

Answer; Following reviewer’s comment, the average CTC number (control vs FFX); is as follows; 19.3 vs 78.1 in 1-2 weeks (p<0.05), 3.5 vs 19.9 in 3-4 weeks (p<0.05) and 31.1 vs 23.5 in 5-7 weeks (p>0.1). This is because CTC number of control and FFX treated mice in 5-7 weeks has large variation with high SD value.

  1. While the number of circulating tumor cells is important, the expression levels of mutant genes or specific mRNA genes in the separated cells are also critical issues. It would be beneficial to add content on this topic to the Discussion section.

Answer; According to the reviewer’s comment, we added following sentences in  Conclusions section as follows; In addition to such a role of CTC number for an important biomarker, further genetic analysis such as the expression levels of mutant genes or specific mRNA genes in the separated CTCs collected in our platform is also critical issue of liquid biopsy.

  1. The CTC separation method is based on filtration, which is greatly influenced by the size of the CTCs. It would be good to add content on this topic to the Discussion section as well.

Answer; In this study, enrichment of CTCs was performed using our original automated and highly efficient CTC collection apparatus with a 3-dimensional metal filter (8 um pore) chip in combination with a constant fluid pressure system, allowing morphologically damage-less collection of CTCs. I think, metal filter with 8 um pore can capture almost all malignant epithelial cells of our concern. Since mechanistic aspect of our filtration-based CTC collecting platform used in this study is previously described in detail (Reff;21). In this study, we added some comments described above in the 2.6 subsection of Materials and method section.

Reviewer 3 Report

Comments and Suggestions for Authors

Here, the authors present an article focusing on  the mobilization of circulating tumor cells (CTCs) after chemotherapy in mouse models of human pancreatic cancer. The authors used two pancreatic cancer cell lines (SUIT-2 and KP-4) to create xenograft mouse models and examined the changes in CTC numbers and primary tumor histology after single-dose and repetitive-dose chemotherapy with FOLFIRINOX (FFX) or Gemcitabine and nab-Paclitaxel (GnP). The articles showed that CTC numbers transiently increased 1-2 weeks after chemotherapy, regardless of the drug regimen and cell line, and correlated with the anti-tumor activity of the drugs. They also observed severe growth arrest, apoptosis, and blood vessel disruption in the primary tumors during this period, suggesting that CTC mobilization was driven by the shedding of degenerated tumor cells into the damaged blood vessels. Finally, the authors proposed that sequential CTC monitoring during chemotherapy could be a useful liquid biopsy diagnostic tool to predict tumor chemosensitivity and resistance in preclinical and clinical settings.

Main Points:

The article provides novel insights into the dynamics and mechanisms of CTC mobilization after chemotherapy in pancreatic cancer, which is very relevant for the readership of the journal.

The article uses two different cell lines and two different drug regimens to demonstrate the consistency and robustness of the CTC mobilization phenomenon across different experimental conditions.

The article uses a novel microfluidic platform and immunocytochemistry to detect and characterize CTCs, which are sensitive and specific methods that can capture both single and cluster CTCs. This ads to the novelty of the article.

Minor Points:

The article uses subcutaneous xenograft models, which may not fully mimic the orthotopic and metastatic features of pancreatic cancer in humans, that also involve immunological responses. Could the authors discuss this study limitation?

The article does not discuss the biological and clinical significance of the CTCs mobilized after chemotherapy, such as their viability, phenotype, and metastatic potential. Could this be discussed?

The article does not compare the CTC mobilization with other biomarkers, such as circulating tumor DNA or exosomes, which may also reflect the tumor response to chemotherapy. Is there a link between high CTC mobilization, and release of circulating tumor DNA in the blood?

Comments on the Quality of English Language

English fine, minor typos. 

Author Response

Thank you for the helpful comments to clarify and strengthen our manuscript.

We answer these questions one by one as follows.

Reviewer 3 comment

Main Points:

  1. The article provides novel insights into the dynamics and mechanisms of CTC mobilization after chemotherapy in pancreatic cancer, which is very relevant for the readership of the journal.
  2. The article uses two different cell lines and two different drug regimens to demonstrate the consistency and robustness of the CTC mobilization phenomenon across different experimental conditions.
  3. The article uses a novel microfluidic platform and immunocytochemistry to detect and characterize CTCs, which are sensitive and specific methods that can capture both single and cluster CTCs. This adds to the novelty of the article.

Minor Points:

1.1 The article uses subcutaneous xenograft models, which may not fully mimic the orthotopic and metastatic features of pancreatic cancer in humans, that also involve immunological responses. Could the authors discuss this study limitation?

Answer; We previously compared subcutaneous and orthotopic xenograft models (Reff. 16). Orthotopic model of pancreatic cancer has several biological advantages including immunological responses, but needs intraperitoneal implantation, which results in increasing surgical stress and risk for fatal peritoneal metastasis in some cases. Therefore, orthotopic model cannot be applicable for long term experiment of the present study. We add this short comment in the 2.4 Animal experiment …of Materials and Method section.

1.2 The article does not discuss the biological and clinical significance of the CTCs mobilized after chemotherapy, such as their viability, phenotype, and metastatic potential. Could this be discussed?

Answer; These are important questions. CTCs detected in this study are judged as viable based on immunocytological findings. Phenotypic expression and metastatic potential of CTC are major biological subjects in itself, but cannot be included in this first study. We are now going to study the relationship between transient increase of CTCs after chemotherapy and metastatic potential. We add such comment in the Conclusion section.

  1. The article does not compare the CTC mobilization with other biomarkers, such as circulating tumor DNA or exosomes, which may also reflect the tumor response to chemotherapy. Is there a link between high CTC mobilization, and release of circulating tumor DNA in the blood?

Answer; This is an interesting subject, but in our mouse model, ctDNA measurement is difficult because of small amount blood sampling (0.2 ml). In the present study, CTC enumeration is the only things to be able to do. Molecular analysis of CTC such as comparison with ctDNA is another study in the near future.

Reviewer 4 Report

Comments and Suggestions for Authors

In this manuscript, the author investigates the dynamics of circulating tumor cells (CTCs) post-chemotherapy in terms of number and morphology and the implications of these findings on understanding tumor chemosensitivity and resistance. This paper offers insights into predicting the response of the primary tumor to therapeutic treatment.

Some minor suggestions are raised for your consideration:

1.       In some charts of figures 2-4, the sample size is not clearly mentioned.

2.       There appears to be some inconsistency in figure labeling. Consider revising labels like “No” of mitotic cells/HPF.

3.       The white arrows used in the Hematoxylin-Eosin (HE) staining figures are somewhat hard to discern.

Author Response

Thank you for the helpful comments to clarify and strengthen our manuscript.

We answer these questions one by one as follows.

Reviewer 4 comment

Minor suggestions

  1. In some charts of figures 2-4, the sample size is not clearly mentioned.

Answer; According to the reviewer’s comment, we added description of the sample size (number of mice examined and numbers of high power fields counted for mitotic index) in the 2.8 of the Materials and Method section. Results section and Figure legends.   

  1. There appears to be some inconsistency in figure labeling. Consider revising labels like “No” of mitotic cells/HPF.

Answer; According to the reviewer’s comment, we corrected figure labeling using unified terms.

  1. The white arrows used in the Hematoxylin-Eosin (HE) staining figures are somewhat hard to discern.

Answer; Following reviewer’s comment, we changed white arrows to black arrow heads in Figure 2-4 and black arrows in Figure 6 to discern objects more clearly.

Round 2

Reviewer 2 Report

Comments and Suggestions for Authors

As a result of reviewing the revised manuscript, I conclude to accept the paper.